# Chronic Sufferers and Environmental Conditions

**Sotiria Vrouva** [1,2,*], **Varvara Sopidou** [3], **Emmanouil Sifakis** [2], **Ilias Ntoulaveris** [2], **Georgios Papamarkos** [1], **Gesthimani Tse** [2], **Konstantinos Chanopoulos** [4] **and George Koumantakis** [1]

[1] Physiotherapy Department, School of Health and Care Sciences, University of West Attica (UNIWA), 12243 Athens, Greece; g.papamarkos@hotmail.com (G.P.); gkoumantakis@uniwa.gr (G.K.)

[2] Department of Physical Therapy, 401 Army General Hospital of Athens, 11525 Athens, Greece; msephakes@gmail.com (E.S.); ilias.ntoulaveris@gmail.com (I.N.); mania.tse@hotmail.com (G.T.)

[3] Department of Biomedical Sciences, School of Health and Care Sciences, University of West Attica (UNIWA), 12243 Athens, Greece; vsopidou@uniwa.gr

[4] Department of Application Developments, Hellenic Military Geographical Service (HMGS), 11362 Athens, Greece; k_hanos@yahoo.com

\* Correspondence: svrouva@uniwa.gr or s.d.vrouva@army.gr

**Abstract:** Environmental conditions are associated with the onset of pain or the recurrence of symptoms of chronic sufferers either with musculoskeletal pathologies or neurological diseases. Recent research has indicated that physiotherapy also appears to be helpful in dealing with the symptoms of climate change. The purpose of this study was to investigate how temperature and precipitation are associated with referrals for physical therapy. All the data were collected retrospectively for three years, 2020–2022. A total of 2164 referrals were studied, 78% of which were female cases. Our results highlighted that (a) referrals with musculoskeletal problems are associated with the weather indicators of temperature and precipitation, (b) this relation was found to be stronger for females, and (c) there were slightly differentiated trends between them and those with neurological problems. These results suggest that low temperatures and high precipitation are strongly associated with increased referrals for chronic musculoskeletal pain and that females are more vulnerable to precipitation. Moreover, the increased number of referrals with chronic neurological problems was found to be associated with extreme temperatures. Considering these findings, rehabilitation centers and healthcare systems have the opportunity to immediately provide reliable and qualitative services, guided always by the safety and maximum relief of chronic sufferers.

**Keywords:** chronic musculoskeletal pain; neurological disease; weather; temperature; precipitation

## 1. Introduction

In recent years, there has been constant talk about weather conditions and the places that affect people's lives and health [1–6]. Environmental changes, such as increases in the intensity of ultraviolet radiation, ionization and abrupt meteorological changes in temperature, wind, and humidity, often lead to difficulties in the adaptation of the urban population [2,4,5,7].

Many researchers have tried to examine whether there is a relationship between the weather and the onset of pain or the recurrence of symptoms of chronic sufferers [1]. Although conducted in different environments (tropical, northern, or humid) with dissimilar assessment tools and subjects and small or large population samples, the relationship between the occurrence of pain and the feeling of discomfort appears to be moderated by temperature, precipitation, and barometric pressure [2,6,8]. By understanding the relationship with these weather variables, it would be easier to delve into the mechanisms that produce pain and control it [7]. After all, the influence of the endocrine system, the effect on our thermoregulatory system, and the change in the circulation, blood viscosity, and heart rate could be the scientific explanations for all the previous problems [7,8].

Meteoropathy, the special sensitivity of some people to weather conditions and the difficulty in managing it, is common [4,8,9]. Almost a third of the population is weather sensitive [7]. These are usually chronic sufferers either with musculoskeletal problems (such as fibromyalgia, seronegative arthritis, or osteoarthritis of the limbs, etc.) [9–12] or chronic neurological diseases such as multiple sclerosis, stroke, and Parkinson's disease, etc. [13–15]. The malfunction is mainly found in adaptation to very high or very low temperatures and sudden changes in atmospheric pressure and humidity [5,6,12]. In addition to discomfort and persistent pain, some patients experience an exacerbation of migraine vertigo, as well as psychosomatic symptoms [12].

Patients frequently turn to rehabilitation units for weather-related pain and range of motion [16]. These chronic sufferers are often referred for physical therapy in order to be more flexible in the changes they observe in their body, with interventions to retrain motor control, behavior changes, and reductions in fear of movement [15]. Physiotherapy also appears to be helpful in dealing with the symptoms of climate change [16]. People with neuromuscular diseases affected by increases in temperature are looking for a solution to the problems of weakness and fatigue in functionality and everyday tasks [17]. The question that arises is whether referrals for physical therapy are weather-related. To our knowledge, there is no previous study examining the potential association of physical therapy referrals of patients suffering from chronic neurological problems with monthly temperatures and precipitation. The aim of the study was to investigate the association of average monthly temperatures and precipitation with the number of patients suffering and the worsening of their symptoms to the point that required a hospital referral. The secondary objective was to explore if this effect was differentiated according to the patients' gender.

## 2. Materials and Methods

This was a retrospective study. After approval by the Institutional Review Board of 401 Military Hospital, four physical therapists independent of the research collected the data. The material was gathered from the records maintained in the Department of Physiotherapy of the 401 Army General Hospital in Athens and was related to the number of patients suffering from chronic pain referred to it, per month for the years 2020–2022. Weather data collected from Acharnon weather station (ELEV: 210 m LAT: 38°06′00″ N LONG: 23°42′00″ E) in Athens consisted of temperature in degrees of Celsius and precipitation produced by rain, snow, sleet, or hail that falls to the ground, in millimeters. Precipitation occurs when a portion of the atmosphere becomes saturated with water vapor (reaching 100% relative humidity), so the water condenses and "precipitates" or falls. The mean monthly temperature and precipitation level were calculated as averages over a 24 h period. A total of 2164 referrals were studied. There were more female cases (1681 or 78%) than male (482, or 22%). Their mean age was 49 ± 13 years old and all of them consisted of military personnel working in various positions with different specialties, with most of them spending enough time of their work outdoors. Chronic pain sufferers referred to the department were categorized into those with chronic musculoskeletal pain such as fibromyalgia, arthritis, chronic shoulder pain, and lower back pain and into those with chronic neurological problems such as Parkinson's disease and multiple sclerosis. In total, 937 patients (169 males and 768 females) with chronic musculoskeletal pain and 1227 (313 males and 913 females) with chronic neurological problems were eligible for this analysis. The study involved referrals for physical therapy and therefore written informed consent was not sought from the patients.

## 3. Statistical Analysis

All data under consideration were tested for normality with the Shapiro–Wilk test. First, a correlation analysis was conducted to discover the relationships between the monthly referrals to the physiotherapy and weather conditions, monthly temperatures, and precipitation levels. We used Pearson's correlation coefficient for normally distributed

data and Spearman's correlation test for the data which seemed not to fit well to normal distribution.

Then, multivariate linear regression analyses were performed to identify the independent weather factors associated with the chronic sufferers (patients with musculoskeletal pain and/or neurological problems). In particular, we wanted to see if the number of referrals could be predicted by weather indicators. All the key assumptions for the regression were examined: a. linear relationship between the outcome variable (referrals) and the independent variables (temperature and precipitation) tested with scatterplots, b. no multicollinearity of the independent variables examined with correlation coefficients (values between 0.3 and 0.8), c. the homogeneity of residual of variances (homoscedasticity) justified with a visual inspection of the standardized residuals versus the predicted values scatterplot, d. the normal distribution of the residuals checked visually using Q-Q plots, and e. the independence of the residuals (and thus the observations) examined with the Durbin–Watson test (value between 1.5 and 2.5). Regression analyses were also conducted exclusively on the subgroups of patients with chronic musculoskeletal pain based on their gender, in order to observe if they were affected differently by weather conditions.

In addition, the data were examined for seasonality and stationarity and a time series analysis was performed with moving averages used to identify trends by smoothing out short-term fluctuations in the data and highlighting longer-term patterns. A trend analysis was conducted to identify patterns and trends in the referrals over time, indicating the direction and strength of this trend. The "decompose" function in R was used to estimate the trend and seasonal components of the number of referrals as a seasonal time series that can be described using an additive model. The additive model used was:

$$Y_t = T_t + S_t + e_t$$

The function first determines the trend component using a moving average and removes it from the time series. Then, the seasonal figure is computed through averaging for each time unit over all periods. The seasonal figure is then centered. Finally, the error component is determined by removing the trend and seasonal figure (recycled as needed) from the original time series. Statistical analyses were performed with SPSS version 24 software (SPSS, Inc., Chicago, IL, USA) and the significance level was set at $p < 0.05$. Descriptive statistics are presented as mean $\pm$ SD (standard deviation). Time series plots and estimated seasonal factors were designed and calculated using R software (version 4.2).

Concerning the analysis performed, there were some potential limitations. There were several extraneous variables (such as barometric and air pollutants) which were not investigated in this survey that could potentially affect the outcomes of our research study. Furthermore, although most of our patients consisted of employees spending enough time outdoors and the effect of climatic conditions was assumed to be immediate, it is common knowledge that environmental stressors often show effects that are delayed in time, requiring the use of statistical models that are flexible enough to describe the additional time dimension of the exposure–response relationship. These could potentially impact the reliability and generalizability of our findings.

## 4. Results

During the last three years, from January 2020 until December 2022, the monthly average temperature and precipitation were found to be 18.16 $\pm$ 7.05 °C and 35.59 $\pm$ 33.29 mm, respectively. A monthly average of 26.03 $\pm$ 25.75 referrals due to musculoskeletal problems was found (4.69 $\pm$ 5.03 for males and 21.33 $\pm$ 21.54 for females), but for patients referred due to neurological problems, this was found slightly higher 34.08 $\pm$ 14.81 (8.69 $\pm$ 5.07 for males and 25.39 $\pm$ 11.15 for females). The mean values and standard deviation of the study variables for each year are presented in Table 1.

**Table 1.** Descriptive characteristics of the study variables.

| Year | | 2020 | | 2021 | | 2022 | |
|---|---|---|---|---|---|---|---|
| | | **Mean** | **SD** | **Mean** | **SD** | **Mean** | **SD** |
| Monthly temperatures (°C) | | 18.2 | 7.09 | 18.09 | 7.16 | 18.2 | 7.50 |
| Monthly precipitation level (mm) | | 36.33 | 34.37 | 37.64 | 35.57 | 32.79 | 32.66 |
| Monthly referrals with musculoskeletal problems | | 24.42 | 25.43 | 28.50 | 26.10 | 25.17 | 27.80 |
| | Male | 5.0 | 4.5 | 4.58 | 5.1 | 4.50 | 5.7 |
| | Female | 19.42 | 21.59 | 23.92 | 22.28 | 20.67 | 22.39 |
| Monthly referrals with neurological problems | | 33.08 | 14.9 | 34.42 | 15.24 | 34.75 | 15.56 |
| | Male | 7.92 | 4.6 | 8.42 | 5.79 | 9.75 | 5.03 |
| | Female | 25.17 | 12.08 | 26.0 | 11.29 | 25.0 | 11.04 |

SD = standard deviation.

Although the Shapiro–Wilk test showed that all data regarding the referrals due to chronic musculoskeletal problems fit to a normal distribution, the number of referrals due to neurological problems appeared not to be normally distributed and Spearman's, the nonparametric correlation coefficient, was used.

Spearman's correlation coefficients revealed strong associations among the weather indicators and total monthly referrals due to chronic musculoskeletal pain. A significant negative moderate relation was found for the number of total referrals with monthly temperatures ($r_s = -0.744$, $p < 0.001$) and a strong positive one with precipitation ($r_s = 0.894$, $p < 0.001$). Of course, high temperatures were found to be correlated with a low level of participation ($r_s = -0.655$, $p < 0.001$). No significant correlations were found for those referred for physical therapy due to neurological disorders.

Since the key assumptions of the multivariate linear regression for referrals with musculoskeletal problems were satisfied, we present below the findings of this analysis. A multivariate linear regression model with dependent variables of the total number of referrals with musculoskeletal problems and predictors, the temperature and level of precipitation revealed a statistically significant correlation between them ($b = -0.763$, $p < 0.025$ for temperature and $b = 0.598$, $p < 0.001$ for precipitation). The model demonstrated 83.3% of the total variance (R square = 0.833). Specifically, we found a 0.763 decrease ($\pm 0.324$) in the total referrals' number for every one-unit increase in temperature and a 0.598 increase ($\pm 0.069$) for every one-unit increase in precipitation. When the model was applied exclusively for subgroups, we found that precipitation associated with female referrals was stronger than with male referrals ($b = 0.509$, $p < 0.001$ and $b = 0.089$, $p < 0.001$, respectively). Temperature was found to be significantly negatively associated with female referrals only ($b = -0.567$, $p < 0.047$), but not with male referrals ($b = -0.196$, $p = 0.05$) (Table 2). On the contrary, the model with the dependent variable of the total number of referrals due to neurological problems was found not to fit well and only a small proportion of the total variance was explained (R square = 0.045).

The results of time series analysis of the monthly referrals for physical therapy revealed differentiated trends between referrals with musculoskeletal problems and those with neurological problems. By decomposing the time series of total referrals with musculoskeletal problems, we managed to estimate the trend; there were seasonal and irregular components (Figure 1).

**Table 2.** Multivariate regression models of monthly referrals with musculoskeletal problems and weather conditions.

| Model | | Unstandardized Coefficients | | 95% Confidence Interval for B | | |
|---|---|---|---|---|---|---|
| | | B | Std. Error | Lower Bound | Upper Bound | *p* Value |
| a. Dependent Variable: Total Monthly Referrals due to musculoskeletal problems | (Constant) | 18.612 | 7.812 | 2.719 | 34.505 | 0.023 |
| | Average Monthly Temperature | **−0.763** | **0.324** | **−1.422** | **−0.103** | **0.025** |
| | Average Monthly Precipitation | **0.598** | **0.069** | **0.458** | **0.737** | **0.000** |
| b. Dependent Variable: Male Monthly Referrals due to musculoskeletal problems | (Constant) | 5.084 | 2.317 | 0.369 | 9.798 | 0.035 |
| | Average Monthly Temperature | −0.196 | 0.096 | −0.391 | 0.000 | 0.050 |
| | Average Monthly Precipitation | **0.089** | **0.02** | **0.048** | **0.130** | **0.000** |
| c. Dependent Variable: Female Monthly Referrals due to musculoskeletal problems | (Constant) | 13.529 | 6.636 | 0.028 | 27.03 | 0.050 |
| | Average Monthly Temperature | **−0.567** | **0.275** | **−1.128** | **−0.007** | **0.047** |
| | Average Monthly Precipitation | **0.509** | **0.058** | **0.39** | **0.627** | **0.000** |

The coefficient indicates the magnitude of change in total referrals expected from a 1-unit change in the independent variable.

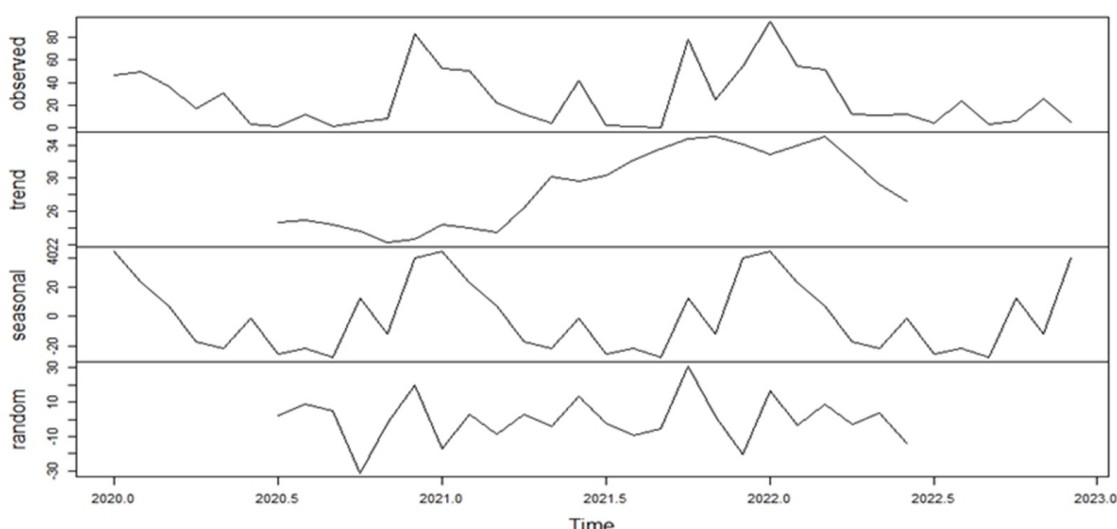

**Figure 1.** Original time series (observed), the estimated trend, seasonal, and irregular (random) components of total monthly referrals with chronic musculoskeletal problems analyzed as time series.

The estimated trend component showed a small decrease from about 26 to about 22 in the second half of 2020, followed by a steady increase from then on to about 34 at the end of year 2021. Then, a sharp decrease to 28 occurred in the first half of 2022. Respectively,

the decomposition of the time series of total referrals with neurological problems (Figure 2) revealed that the estimated trend component showed a small but steady increase from about 33 in the first half of 2020 to about 37 at the start of the year 2022, followed by a smooth decrease until the end of the first half of 2022.

**Decomposition of additive time series**

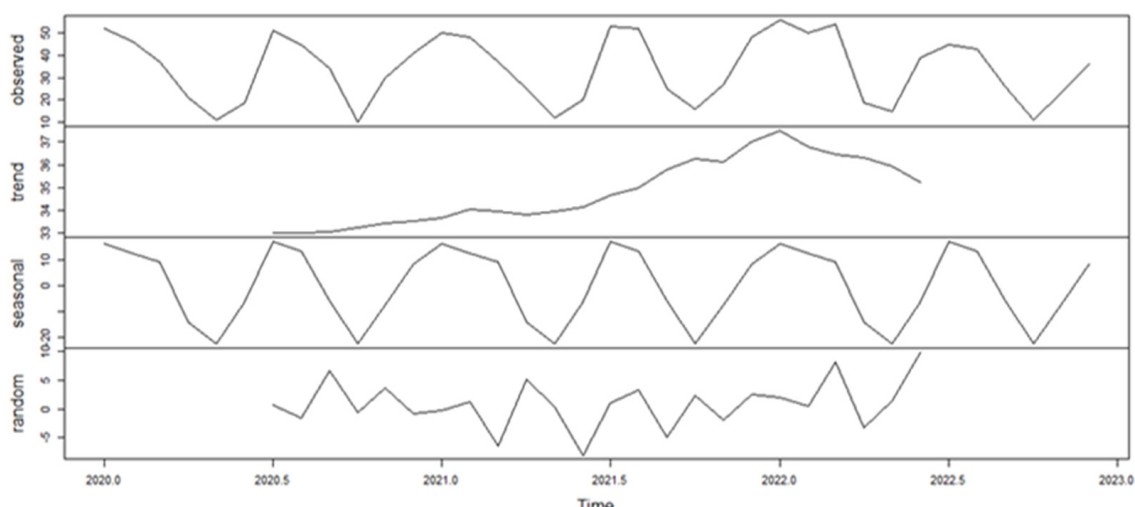

**Figure 2.** Original time series (observed), the estimated trend, seasonal, and irregular (random) components of total monthly referrals with chronic neurological problems analyzed as time series.

The estimated seasonal factors for the referrals due to musculoskeletal problems were computed for the months of January–December and the same for was found each year, which indicates seasonality. The largest seasonal factor occurred in January (about 44.56), and the lowest occurred in September (about −28.25), indicating that there seemed to be a peak in referrals in January and a trough in September each year. Concerning referrals with neurological problems, the largest seasonal factors occurred in July and January (about 17.29 and 16.54, respectively), and the lowest occurred in October and May (about −22.62 and −22.31, respectively), indicating the peaks and troughs in referrals in these months each year.

Moreover, monthly box plots for the study variables clearly show that the number of total referrals with musculoskeletal problems increased in those months with high levels of precipitation and low temperatures (Figure 3). Specifically, there were more than 45 referrals in cold months with precipitation levels of up to 40 mm (December, January, and February), whereas there were less than five referrals in warm months with very low precipitation levels (July, September).

On the other hand, the number of total referrals with neurological problems appeared to be increased in these months with excessively high and low temperatures, while it was not found to be affected by precipitation levels. Specifically, there were more than 45 patient referrals in the coldest months (January and February) with low temperatures at about 10 °C, but also in the hottest months of the year (July and August) with temperatures more than 25 °C (Figure 3). The lowest number of referred patients, at about 10, occurred in months with the temperature ranging from 18 to 23 °C (May and October).

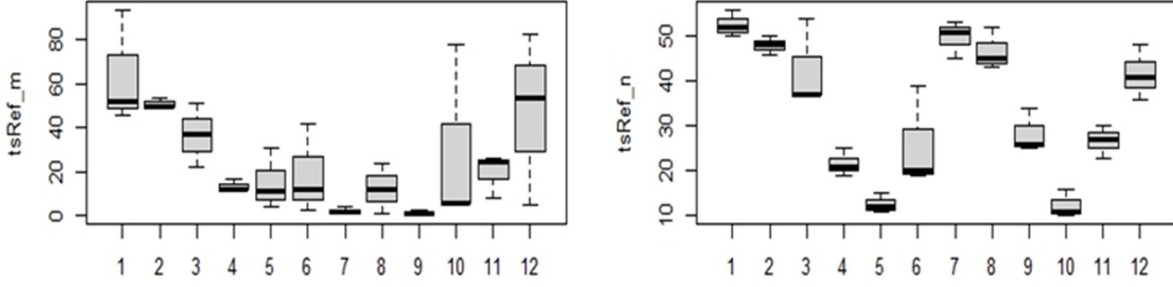

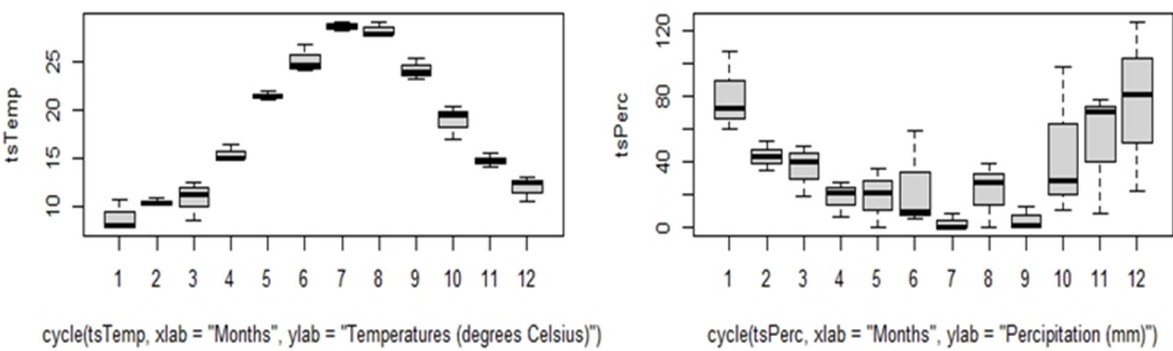

**Figure 3.** Monthly box plots for the study variables analyzed as time series for three years (2020–2022).

## 5. Discussion

The specific meteorological station was chosen because the patients referred to the department all came from regions of Attica (Athens, Greece). Since the prefecture of Attica includes areas close to the sea but also more mountainous, it was decided to make this choice in order to have an average altitude.

### 5.1. Seasonal Variation

Human organisms, through the mechanisms of homeostasis, try to compensate for temperature changes; however, this is limited or not applicable when reference is made to the elderly or patients [18,19]. Several studies have agreed that pain in musculoskeletal patients exhibits seasonality [2,9,20]. Most have agreed to the fact that pain worsens in spring and winter [20], while others have reported that the number of referrals is greater in the period from October to May than during summer [7]. In our research, referrals increased during the months of December, January, February, and March, i.e., in spring and winter. Some authors in the past have attempted to investigate the relationship between weather conditions and the onset or worsening of pain [21].

### 5.2. Mechanisms in Chronic Disease Sufferers

Not all meteor-sensitive individuals can develop meteoropathy [22]. In chronic sufferers, central sensitization may not only affect the perception of pain, but also be reinforced by other non-painful stimuli [23]. However, up to 97% of patients with chronic musculoskeletal pain believe that rain and atmospheric pressure affect them [1]. Under low atmospheric pressure levels, the fluids and air in the inflamed joints expand, causing an increase in pain [1]. It is also possible that low barometric pressure activates the body's baroreceptors [15]. Often, an increase in barometric pressure or a change in ambient temperature, as well as increased humidity, cause a decrease in the path of pain and changes in the viscosity of synovial fluid, reducing its production and eliminating the elasticity of the

already affected peri-articular structures [24]. Apart from weight receivers, both low but also high ambient temperatures with a high relative humidity activate the thermoreceptors of the skin with the parallel secretion of inflammatory substances (endothelial growth factor, interleukin-1 [IL-1]), creating or increasing inflammation in the joints [19]. It is widely believed that barometric pressure stimulates vestibular neuronal activation by acting on the hypo-thalamus-pituitary-adrenal axis (HPA axis), causing the release of hormones from the adrenal cortex (cortisol and catecholamines) [19] to cause vasoconstriction, thus increasing pain [25]. Seasonal changes in vitamin D levels may affect the expression of this action [9]. Shultzet al. (2020) [9] reported that, on an app recording the data of more than 10,000 people with musculoskeletal conditions, 20% experienced an increase in pain when precipitation increased.

Other studies have reported that temperature has no significant correlation with the onset of pain, but is mainly associated with an increase in humidity of at least nine units [26]. In general, an increase in temperature of >1.66 °C or an increase in humidity above 70% leads to pain differentiation in patients with osteoarthritis or fractures up to one year after the event [26]. A decrease in temperature causes vasoconstriction and causes the brain to feel threatened [7]. On the contrary, increasing the temperature or humidity to more than 70% causes vasodilation for thermoregulatory reasons [26]. Furthermore, in patients with musculoskeletal problems due to inflammatory mediators, pain fibers are already affected and changes in temperature and pressure are converted into a painful stimulus [19]. Moreover, changes in electromagnetic fields such as those occurring in thunderstorms can alter the nerve membrane potential, leading to enhanced permeability to different ions that further enhancing the sensation of pain [27].

### 5.3. Chronic Musculoskeletal Pain

Low temperatures and high precipitation rates are associated with an increased number of referrals with chronic musculoskeletal pain. According to Beukenhorsta et al. [1], 24 of the 43 studies showed a positive correlation between pain, temperature, and level of precipitation. This report is related to the diseases of rheumatoid arthritis, osteoarthritis, fibromyalgia, lower back pain, gout, osteoarthritis, spondyloarthritis, arthritis, and adhesive capsulitis of the shoulder [1]. It was emphasized, however, that the key to understanding this correlation is to consider exposure times in environments that exhibit these characteristics [1]. Moreover, the urban populations have lost their ability to adapt to changes in weather [7,13]. Weather conditions affect the secretion of adreno-corticotropic hormones, reduce endorphin levels, and therefore increase the sensation of pain [3]. Temperature and humidity levels inside buildings differ from those in the outside environment [7,8,13]. For this reason, there are differences in their effect on people working indoors [7]. Outdoor exposure for more than half an hour is capable of causing symptoms in weather-sensitive individuals [7]. Since the sample of chronic musculoskeletal pain in our study consisted of military personnel working for quite a long time outdoors, the effect of these climatic conditions was assumed to be immediate [7]. One third of the population, including people of all ages (even children), suffer from meteoropathy [7]. Women comprise the largest percentage, especially those who are menopausal [7]. Similarly, in this research, it appeared that females with musculoskeletal problems were more affected by temperature than males, while high levels of humidity were shown to be related only to female referrals. The different body composition levels between the sexes probably explain why middle-aged women are more sensitive to weather changes than men below 40 and under 66 years old [28].

### 5.4. Neurological Diseases

For neurological referrals, things seem to be different. The increased number of referrals with chronic neurological problems was found to be associated with extreme temperatures (low or high) and not with precipitation. In a study that focused on the period of 1990–2016 and explored how many were affected by the intense warming in several

countries, those affected presented higher increases in Parkinson's disease indicators (such as prevalence, deaths, and disability-adjusted life) [18]. Rowell R (2017) [29], in his research, reported that temperature can affect the symptoms of Parkinson's disease. Since the brain is recognized as the body's most heat-sensitive organ [30], hot weather may cause more PD patients to experience increased symptoms, resulting in worsening tremors, bradykinesia, a lack of balance, depression, and cognitive dysfunctions [29]. In fact, he added that Australian doctors should seasonally adjust prescriptions. The prescribed medication was 4.2% greater in January and 4.5% lower in July [29]. Similarly, we found that peaks in the number of referrals for physical therapy due to neurological problems occurred in these months, January and July, which demonstrates extreme low (<10 °C) or extreme high (>25 °C) temperatures. Excessive exposure to heat could lead to a greater predisposition to worsening neurodegenerative disease [30]. The increasingly higher temperature of the environment due to global warming also causes a prolonged acclimatization process [29]. However, in this way, the oxidizing agent is activated stress and the excitotoxicity that brings about additional neurodegeneration [31].

In patients with multiple sclerosis, changes in body temperature, as they arise from the interaction of body temperature and the environment, lead to difficulties in daily movements. Motor, sensory, and cognitive functions often deteriorate [32]. In the majority of patients, there are increases in spasticity, gait disturbances, and lack of balance, even at a temperature increase of 0.5 degrees [33]. Similarly, in cold environments, almost half of Parkinson's patients develop an extreme sensitivity in their limbs that worsens the problem of standing and walking as the pain occurs due to the cold [34]. It is recalled that these individuals have a low temperature in the lower extremities, even in summer [34]. However, it is not just hot environments that affect MS patients. Temperatures below 20 °C also create problems [32]. Thus, it is common for them to exhibit mechanical as well as cold allodynia [35]. A total of 5% of patients report sensitivity to cold during cold baths. In the winter, there is a clear worsening of symptoms and quality-of-life [32]. Although the occurrence of strokes is associated with low temperatures, leading to an increase in blood pressure [36], the literature does not have sufficient evidence on whether pain and spasticity in hemiplegics are related to weather conditions.

*5.5. Implications*

The correlation between the worsening of musculoskeletal pathologies and variations in temperature, pressure, and environmental humidity could be exploited to design new therapeutic approaches that take into account environmental and barothermal aspects in the administration of rehabilitation treatments. However, variations in patient load that are expected to occur due to environmental factors can be assessed by healthcare facilities in order to organize appropriately to meet these needs. In addition to prevention, the findings of this study could guide more effective clinical care. Some researchers report that environmental conditions affect access to physical therapy services [37]. They describe that, although chronic cold or heat sufferers potentially need more physical therapy services, their desire to do so is often inhibited by the weather. The rise in temperatures above 23 degrees Celsius makes it easier for patients to move around and increases the demand for physical therapy treatment. In our research, the corresponding issue was not observed, given that temperature changes are not accompanied by extreme weather phenomena [24]. Palstam et al. (2021) [37] reported that the value of physical therapy is a function of effectiveness in patients versus the sum of environmental, social, and psychological factors. In conclusion, you should take into account the influence of the environment because, as it increases, the effectiveness of the therapeutic rehabilitation decreases [37].

## 6. Limitations

The research findings came from the measurements of a single weather station. This in itself is a limitation despite the fact that it was chosen according to average altitude. To ensure the accuracy of the recording of weather conditions in a very large area such as

AtticaRegion, several meteorological stations located near the place of residence or work of each sufferer should have been used as sources. However, this was not workable. Moreover, due to the COVID-19 pandemic and the implementation of teleworking, we do not know if the requests for access to rehabilitation care would have been more without the related restrictions. Given the structured nature of our survey, our ability to probe deeper into individual responses is limited. This means we may not fully understand the context or reasoning behind the referrals, potentially limiting the depth of our findings.

Because of the COVID-19 pandemic occurring at the same time that data collection took place in this study, we suggest collecting more relevant data from primary sources, which could provide more accurate and timely insights into this phenomenon for future research. Furthermore, because of the non-linear and lagged effects of some weather data on chronic sufferers pain, we suggest that future studies should use generalized linear regression combined with a distributed lag non-linear model (DLNM) to compare and contrast the results among different diseases, after controlling for long-term trends and seasonality. DLNM can simultaneously investigate the non-linear exposure–response relationships and delayed effects in a flexible way, and it has been widely used to examine the relationships and delayed effects of meteorological variables on human health in various diseases [37–40].

## 7. Conclusions

Low temperatures and high precipitation rates were found to be associated with an increased number of referrals of patients with chronic musculoskeletal problems. It was also demonstrated that females were more vulnerable to precipitation increases than males. Moreover, the increased number of patients referred with chronic neurological problems was found to be associated with extreme temperatures (low or high) and not with precipitation. The findings of this study should alert all healthcare stakeholders, from hospital managers to referring clinicians and therapy providers, of the fluctuations in patient caseload that are expected to occur due to environmental factors, in order to enable them to maintain the quality of services provided. Furthermore, taking into account weather conditions, as well as their relationship with the number of referrals to rehabilitation centers due to musculoskeletal or neurological diseases, it enables them to immediately provide reliable services, always guided by the safety and maximum relief of chronic sufferers.

**Author Contributions:** S.V., conceptualization, methodology, writing—original draft; V.S., conceptualization, methodology, writing—original draft; E.S. and I.N., investigation, G.T., data elaboration; G.P., data, elaboration; K.C., statistical analysis, G.K., visualization, review, and editing. All authors have read and agreed to the published version of the manuscript.

**Funding:** This research received no external funding.

**Institutional Review Board Statement:** The study was conducted and approved by the Institutional Review Board of 401 Army Military Hospital (protocol code3/2023, date of approval 5 March 2023). The study involved referrals for physical therapy and therefore written informed consent was not sought from patients.

**Informed Consent Statement:** Not applicable.

**Data Availability Statement:** The data are not publicly available due to privacy restrictions.

**Acknowledgments:** We would like to thank K. Liaskonis and G. Dimitrakopoulos of the 401 Army General Hospital for their support in conducting the research. We also thank N. Rouvela and N. Moschona for their contribution to the statistical analysis.

**Conflicts of Interest:** The authors declare no conflict of interest.

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
