# Peer review of "Chronic Sufferers and Environmental Conditions"

_safety_

Round 1
Reviewer 1 Report
Comments and Suggestions for Authors
The paper is overall well written and quite interesting. The topic addressed is of particular interest to better understand how environmental factors can influence the state of musculoskeletal health of human beings. There is an objective criticality, already reported by you, regarding some limitations of the study (concerning the accuracy of the recording of atmospheric conditions in a very large area and the traceability of requests for access to rehabilitation care) which, however, I do not believe can significantly change the results of the study, given the good size of the sample considered. I would suggest opening up the possibility, in the conclusions section, of considering the results of this study not only from a prevention perspective but also from a treatment perspective, as the analysis of the correlation between worsening of musculoskeletal pathologies and variations in temperature, pressure and environmental humidity could be exploited to design new therapeutic approaches that take into account environmental and barothermal aspects in the administration of rehabilitation treatments.
Reviewer 2 Report
Comments and Suggestions for Authors
Very interesting topic with a very good sample size and statistic analysis, only one question, please add the approval of your ethic comintee, or explain if for a retrospective study is not neccesary in your country
Reviewer 3 Report
Comments and Suggestions for Authors
The relationships between weather conditions and chronic pain have interested me for a long time. I would have liked more information about the participants and where the data collection took place. Where is Attica? You have much more female participants so gender comparison would have been difficult. I would have liked more information about the participant's age, job and what kind of chronic pain they had. In my experience not all chronic pain sufferers feel physical changes due to the weather. I would believe that those with fibromyalgia and arthritis would feel even more than those with low back pain. Where there difference between those working outdoor compared to those working indoors? You mention in the discussion chapter that your sample consisted of military personnel working in the countryside. Why didn't you tell the readers that before? Because of Covid pandemic at the same time that data collection was in this study must have made it difficult to conclude about the influence of the weather.
Reviewer 4 Report
Comments and Suggestions for Authors
Reviewer's Report:
Title: Chronic sufferers and environmental conditions
Abstract:
The abstract provides a brief overview of the study, its purpose, methods, and key findings. However, there are several issues with the abstract:
1. Lack of Clarity: The abstract lacks clarity and conciseness. It should be more succinct and to the point. The information about the number of referrals, the weather indicators, and the strength of the associations could be presented in a more straightforward manner.
2. Data Presentation: The abstract could benefit from presenting the key results in a more organized format, such as a bulleted list or a table, to make it easier for readers to grasp the main findings quickly.
3. Missing Context: The abstract does not provide sufficient context for the study. It mentions the purpose and methods but does not explain why this research is important or what potential implications the findings may have.
Introduction:
The introduction section introduces the general topic of weather conditions and their potential impact on people's health, particularly chronic sufferers. However, there are several issues with this section:
1. Lack of Focus: The introduction lacks a clear focus on the specific research question or hypothesis being addressed in the study. It mentions various aspects of weather and health without narrowing down the scope.
2. Citation: The section mentions the need for a citation, which is missing. Properly citing relevant literature is crucial to support the background information and establish the significance of the research.
3. Organization: The introduction lacks proper organization, making it difficult for readers to follow the logical flow of the information. It should provide a clear roadmap of what the study aims to investigate.
Methods:
The methods section describes the data collection, analysis procedures, and statistical techniques used in the study. However, there are several issues with this section:
1. Lack of Detail: The methods section lacks sufficient detail on how weather data were collected, the specific variables measured, and the sources of data. Providing more information would enhance the reproducibility of the study.
2. Data Presentation: The description of the data analysis procedures is somewhat convoluted. It would be beneficial to present the steps of the analysis in a more structured and straightforward manner.
3. Statistical Analysis: While the section mentions statistical analysis, it does not provide details about the statistical tests used, assumptions made, or any potential limitations of the analysis.
4. Seasonality and Stationarity: The section mentions that data were examined for seasonality and stationarity, but it does not provide any results or insights from this analysis. Including this information would be valuable for understanding the data patterns.
Results:
1. Data Presentation: The presentation of results in the Results section lacks clarity and organization. While the descriptive statistics in Table 1 are informative, the narrative description of the findings could be improved. It would be helpful to summarize the key findings in a more concise manner.
2. Lack of Interpretation: The section provides statistical results but lacks interpretation and discussion of the implications of these findings. It is essential to explain why these results are significant and how they relate to the research question and existing literature.
3. Lack of Visuals: The section mentions time series analysis and decomposition but does not include any visual representations, such as graphs or figures, to illustrate the trends and patterns. Visual aids can enhance the reader's understanding of the data.
Discussion:
1. Lack of Structure: The discussion lacks a clear structure, making it challenging for readers to follow the flow of the argument. It would benefit from organizing the discussion into subsections based on different aspects of the findings.
2. Limited Citations: The discussion references some studies but could benefit from more comprehensive citations to support the claims made. Additionally, the discussion would be strengthened by citing relevant literature throughout to provide context and support for the conclusions.
3. Interpretation of Findings: The discussion should provide a more in-depth interpretation of the results presented in the Results section. For example, it should explain why certain weather conditions affect referrals for musculoskeletal problems but not for neurological problems and what potential mechanisms could underlie these associations.
4. Implications: The discussion should elaborate on the practical implications of the findings. How can healthcare providers and policymakers use this information to improve patient care or service planning? What are the implications for patients with chronic conditions?
5. Limitations: The limitations mentioned in the section are relevant but could be expanded upon to provide a more comprehensive understanding of the study's constraints.
Comments on the Quality of English Language
Minor English editing is required.
Round 2
Reviewer 2 Report
Comments and Suggestions for Authors
excellent biometeorology research, an important physical therapy topic
congratulations
Reviewer 3 Report
Comments and Suggestions for Authors
I have no further comments. Well done and interesting manuscript.
Reviewer 4 Report
Comments and Suggestions for Authors
The authors have conscientiously and successfully dealt with the raised queries, leading to a significant improvement in the manuscript's overall quality. Consequently, I am delighted to recommend the acceptance of the manuscript in its present state.